# Superimposed Mosaicism in the Form of Extremely Extended Segmental Plexiform Neurofibroma Caused by a Novel Pathogenic Variant in the NF1 Gene

**DOI:** 10.3390/ijms241512154

**Published:** 2023-07-29

**Authors:** Klára Veres, Judit Bene, Kinga Hadzsiev, Miklós Garami, Sára Pálla, Rudolf Happle, Márta Medvecz, Zsuzsanna Zsófia Szalai

**Affiliations:** 1Department of Pediatric Dermatology, Heim Pal National Children’s Institute, 1089 Budapest, Hungary; klariii@yahoo.com (K.V.);; 2Department of Medical Genetics, Clinical Center, Medical School, University of Pécs, 7622 Pécs, Hungary; 3Pediatric Center, Faculty of Medicine, Semmelweis University, 1085 Budapest, Hungary; 4Department of Dermatology, Venereology and Dermatooncology, Semmelweis University, 1085 Budapest, Hungary; 5Department of Dermatology, Medical Center–University of Freiburg, 79104 Freiburg, Germany

**Keywords:** neurofibromatosis type-1, neurocutaneous syndrome, whole exome sequencing, segmental plexiform neurofibroma, splice-site mutations, superimposed mosaicism, second hit mutation, loss of heterozygosity, molecular targeted therapy

## Abstract

Plexiform neurofibromas occurring in approximately 20–50% of all neurofibromatosis type-1 (NF1) cases are histologically benign tumors, but they can be fatal due to compression of vital structures or transformation to malignant sarcomas or malignant peripheral nerve sheath tumors. All sizeable plexiform neurofibromas are thought to result from an early second mutation giving rise to a loss of heterozygosity of the *NF1* gene. In this unusual case, a 12-year-old girl presented with a rapidly growing, extremely extensive plexiform neurofibroma with segmental distribution over the entire right arm, extending to the right chest wall and mediastinum, superimposed on classic cutaneous lesions of NF1. After several surgical interventions, the patient was efficiently treated with an oral selective MEK inhibitor, selumetinib, which resulted in a rapid reduction of the tumor volume. Molecular analysis of the *NF1* gene revealed a c.2326-2 A>G splice-site mutation in the clinically unaffected skin, peripheral blood sample, and plexiform neurofibroma, which explains the general clinical symptoms. Furthermore, a novel likely pathogenic variant, c.4933dupC (p.Leu1645Profs*7), has been identified exclusively in the girl’s plexiform neurofibromas. This second-hit mutation can explain the extremely extensive segmental involvement.

## 1. Introduction

Neurofibromatosis type 1 (NF1) is an autosomal dominant neurocutaneous disorder with a worldwide incidence of 1 in 2500–3000 individuals. NF1 is caused by a heterozygous mutation in the tumor suppressor neurofibromin (*NF1*) gene. Due to the large size of the *NF1* gene, which encompasses 350 kb of the genome, it is susceptible to a wide range of mutations, 85–90% of which are point mutations, 5–10% of which are microdeletions, and 2% of which are exon deletions or duplications [1]. The mutation rate of the *NF1* gene is extremely high, and up to 50% of cases are due to de novo mutation [2,3].

The disease shows complete penetrance with age, with 50% of patients demonstrating characteristic clinical signs by the age of 1 year and 97% by the age of 8 years. Diagnostic criteria established in 1987 have been revised in 2021, including six or more café-au-lait macules (CALMs, ≥5 mm in prepubertal individuals); axillary or inguinal freckling; two or more neurofibromas or one plexiform neurofibroma (PN); optic pathway glioma; Lisch nodules or choroidal abnormalities; a distinctive osseous lesion (e.g., sphenoid dysplasia, anterolateral bowing of the tibia, or pseudoarthrosis of a long bone); and a heterozygous pathogenic *NF1* variant [4]. The diagnosis can be established if the patient meets at least two criteria. A child of a parent diagnosed with NF1 should meet only one of the above-described criteria to merit a diagnosis of NF1 [2,5,6,7,8,9,10,11]. Naevus anaemicus, pseudoatrophic macules, glomus tumor, scoliosis, and juvenile xanthogranulomas are also more frequent in NF1 patients [2,4,9,12,13].

Plexiform neurofibromas occur in approximately 20–50% of all NF1 cases [2,14]. Diffuse PNs tend to develop already in early childhood, whereas deep nodular PNs usually appear at a later age [2]. Plexiform neurofibromas are histologically benign tumors consisting of different cell types, including neuronal axons, Schwann cells, fibroblasts, mast cells, macrophages, perineural cells, and extracellular matrix materials, such as collagen, but carry a malignant potential in 5–15% of patients [2,15,16]. They can appear in any part of the body, but most commonly on the face, chest, and lower extremity [7]. The size and severity of the tumor mass increases with age, often causing disfigurement and impaired function of the affected area [2,10,15,17]. The outcome can even be fatal owing to compression of vital structures or transformation into a malignant sarcoma or malignant peripheral nerve sheath tumor [15].

Treating plexiform neurofibromas, which significantly affect quality of life, has long been an unresolved challenge. In the past, they were treated surgically, with numerous operations and only partial results. For smaller neurofibromas, carbon dioxide lasers or electrocautery can be used [2]. Fortunately, new therapeutic and preventive options, including several targeted genetic treatments, have recently become available.

The tumor suppressor protein neurofibromin is encoded by the *NF1* gene. It plays an important role in the Ras/Raf/ERK pathway [2,8,9,13], which is the target of several drugs in the market [18]. Sirolimus, a generally well-tolerated mTOR inhibitor, may delay the progression of PN and decrease the associated pain [2]. Tipifarnib, which blocks RAS signaling by inhibiting farnesylation of RAS, has an impact on quality of life but does not affect tumor progression [2,8,18]. Pirfenidone, an inhibitor of fibroblasts, may inhibit progression, but does not lead to regression of the tumor mass of PN [2,8]. Pegylated interferons decrease pain and may even lead to tumor regression owing to their antiproliferative and antiangiogenic effects [2,8]. Tyrosin kinase inhibitors, such as selumetinib, imatinib, trametinib, mirdametinib, and cabozantinib, have been shown to be the most effective treatment options, leading to 10–20% reduction in tumor volume of PN in clinical trials [2,19,20,21].

## 2. Results

### 2.1. Clinical and Histological Findings

The patient was a 12-year-old female with multiple CALMs, born after an uneventful pregnancy and vaginal delivery. The neonatal presentation, psychomotor development, and mental function were normal. Her family history was positive for CALMs, but negative for other features of neurofibromatosis (Figure 1).

At 5 months of age, yellowish plaques appeared in a segmental distribution on the right arm and right side of the chest, which continuously progressively developed into a large tumor mass causing difficulties in daily life. Several plastic surgery interventions were performed to eliminate the disfigurement and functional impairment.

On physical examination at the age of 12 years, a well-demarcated, brownish-livid soft tumor mass with several mobile, tender nodules was observed on the medial side of the right arm extending from the palm to the axillary region with a segmental distribution that also extended to the right side of the chest (Figure 2). 

The tumor mass on the forearm also showed several scars of previous surgical interventions. Small, soft, skin-colored neurofibromas were detected in the left eyebrow region and on the right hairy scalp in the frontal area. The clinical examination also revealed multiple, homogeneously hyperpigmented macules on the chest and trunk consistent with CALMs (Figure 2B). 

Histopathological specimens taken from the extended segmental lesion on the right forearm revealed a non-encapsulated tumor mass in the dermis consisting of spindle cells with ovoid and wavy bland nuclei with pale eosinophilic cytoplasm. Thickened nerve branches, collagen bundles, and capillaries were detected among the tumor cells. Atypical cell forms were not present. Histopathological findings of the samples were consistent with plexiform neurofibroma (Figure 3). Immunohistochemical testing for the INI1/SMARCB1 has been found mosaic expression, it was positive in the majority of the cells and lost in the smaller part of the cells.

### 2.2. Investigation

As part of the neurofibromatosis investigation, several examinations were undertaken. 

The ophthalmological examination revealed the presence of Lisch nodules. Orthopedic examination revealed scoliosis and pronounced pectus excavatum. Additional findings included mild left ventricular cardial dysfunction, but no treatment was required. Audiology, neurology, abdominal ultrasound, and wrist radiography showed normal results. There were no mental impairments or deviant behaviors. An MRI scan of the chest revealed the formation of multiple nodules corresponding to an extensive plexiform neurofibroma of the right side of the thoracic spine, extending from the lung apex to the upper mediastinum. The lesions were observed between the ribs on the left side of the thoracic eight vertebra and on the right side of the chest wall. Consequential left convex scoliosis was noted in the dorsal segment, with normal straightening of the upper dorsal kyphosis (Figure 4). On the right arm, neurofibromatous soft tissues were depicted medially along the brachial plexus up to the wrist, extending into the muscles and subcutis, with a maximum width of 3.5 cm. Cranial MRI showed no significant intracranial abnormalities.

### 2.3. Orphan Drug Therapy

In April 2020, selumetinib, an oral selective MEK inhibitor that reduces PN size without serious adverse reactions in pediatric patients, was approved by the US Food and Drug Administration for the treatment of children with NF1-related progressive, inoperable plexiform neurofibroma [8,9,18,19,20,21,22]. Selumetinib was initiated for the patient, with significant improvement already after the first cycle of treatment. The whole tumor mass became less tight, the tender nodules decreased from 30 to 10 mm, and her quality of life improved significantly (Figure 5). 

Two months after the administration of selumetinib, an MRI image showed a reduction in size of up to 30% in particular parts of the tumor mass (Figure 6).

According to the follow-up MRI examination, the tumor volume stopped shrinking after approximately 4 months, but no deterioration was observed since then. Clinically, plexiform neurofibroma, localized on the right arm, still undergoes continuous regression. Mild acne, bleaching of hair color, and, after a year, paronychia were observed as side effects. For the latter, the therapy was temporarily suspended twice for two weeks, administration of the drug was restarted, and the patient is currently on selumetinib therapy.

### 2.4. Genetic Analyses

#### 2.4.1. Next Generation Sequencing of *NF1, NF2, RAF1, KIT, SPRED1, SMARCB1, PTPN11* Genes

Pre-test clinical genetic counselling and written informed consent were obtained for molecular genetic investigation. DNA was isolated from the lesion of the forearm and lymphocytes in the peripheral blood. Next generation sequencing (NGS) of the *NF1*, *NF2*, *RAF1*, *KIT*, *SPRED1*, *SMARCB1*, and *PTPN11* genes were performed. A pathogenic splice site mutation (c.2326-2A>G, HGMD Accession Number:CS030307) in the *NF1* gene [NM_001042492.3] was identified in both DNA isolated from both the biopsy sample and peripheral blood (Figure 7). The NGS based gene panel sequencing did not detect any disease-causing mutation in genes *NF2*, *RAF1*, *KIT*, *SPRED1*, *SMARCB1*, and *PTPN11*. The presence of the detected variant was verified by Sanger sequencing by using an ABI3500 Genetic Analyzer (Thermo Fisher Scientific Inc, Waltham, MA, USA).

#### 2.4.2. MLPA Analysis 

Multiplex ligation-dependent probe amplification (MLPA) analysis was performed to check whether the loss of heterozygosity due to deletion of one *NF1* allele can be detected in the plexiform neurofibroma. Loss of heterozygosity is supposed to play a role in the formation of superimposed mosaicism, previously called ‘type 2 segmental mosaicism’ of NF1. The MLPA assay did not reveal any copy number variations either in the plexiform neurofibroma or in the peripheral blood sample. However, it is worth mentioning that MLPA measurements are not capable of detecting low-grade mosaicism below 20% due to the nature of the technique.

#### 2.4.3. Whole Exome Sequencing

Since the MLPA analysis and NGS-based gene panel sequencing did not reveal any differences between the genetic architecture of DNA isolated from peripheral blood and plexiform neurofibroma, a whole exome sequencing was performed on the unaffected skin, peripheral blood sample, and on a second sample of plexiform neurofibroma. During bioinformatic data analysis, sequence reads were filtered for the *NF1* gene. A novel frameshift variant, c.4933dupC (p.Leu1645Profs*7)[NM_001042492.3], has been identified exclusively in the plexiform neurofibroma with a 16% variant allele frequency (VAF) (Figure 8). According to the ACMG classification (PVS1, PM2), this novel mutation can be regarded as likely pathogenic. When a modified bioinformatic pipeline was used for the analysis of NGS gene panel data, this novel variant was detected in the plexiform neurofibroma, too.

## 3. Discussion

The *NF1* gene is located on the gene locus 17q11.2. The gene product protein neurofibromin is a tumor suppressor involved in the downregulation of the RAS signaling pathway. Patients with NF1 lack properly functioning neurofibromin, resulting in uninhibited proliferation of RAS-GTP pathways. There is also upregulation of the mTOR pathway, which promotes the development of malignancies. Neurofibromin is originally present in neurons, oligodendrocytes, and Schwann cells in the postembryonic phase of life, and is also found in other cell types, such as white blood cells, keratinocytes, and adrenal medulla [18]. Neurofibromin is reduced or absent in neurofibroma cells [18,23].

The genetic background of plexiform neurofibromas has extensively been investigated. One *NF1* allele carries a genetic alteration in all cells of a patient with NF1, and a loss of the second *NF1* allele (loss of heterozygosity, LOH) results in a complete loss of neurofibromin function and in tumor development [18,23,24,25,26,27]. De Raedt et al. observed that the proportion of cells exhibiting LOH in dermal neurofibromas varied between 30 and 75% [26,27,28,29]. Upadhyaya et al. found LOH of the *NF1* gene region in around 20% of the cases with benign dermal neurofibromas, more than half (50–70%) of PNFs cases, and in MPNSTs represented greater than 90% of all somatic mutations identified [27]. Moreover, the wide range of tumor subtypes and their diverse locations support the concept of loss of *NF1* at early developmental stages in undifferentiated precursor cells [23]. 

In the present case, a mutation in the *NF1* gene was detected (c.2326-2A>G) in clinically unaffected skin, a peripheral blood sample, and plexiform neurofibroma, which is known to be a pathogenic mutation affecting mRNA splicing [2].

Happle’s theory of superimposed mosaicism may explain the pronounced manifestation of this extensive segmental tumor. According to his hypothesis, superimposed mosaicism, previously called type 2 segmental mosaicism of autosomal dominant disorders, is characterized by a congenital or very early manifestation of the disease in a rather pronounced form [10,11,30,31]. Happle hypothesized that in these cases, an early postzygotic loss of heterozygosity occurs in a heterozygous embryo: the absence of the corresponding normal allele leads to doubling of the mutated genetic burden, resulting in a significant increase in the severity of segmental arrangement [10,30,31]. In our case, the combination of a pronounced plexiform neurofibroma in a segmental distribution superimposed on the classic skin lesions of NF1 could be explained by superimposed mosaicism [31]. The data of molecular investigation corroborated this theory by documenting compound heterozygosity in 16% of the examined cells.

To explore the exact genetic background, whole exome sequencing was performed, which identified a novel likely pathogenic variant in the *NF1* gene (:c.4933dupC (p.Leu1645Profs*7) [NM_001042492.3] ) This theory is reminiscent of, but not identical with, Knudson’s two-hit hypothesis describing mutational events during tumorigenesis [14,32]. All of the benign NF1-associated lesions exhibit a second somatic mutation [14,23]. Pemov et al. investigated 23 NF1-associated PNs and identified germline *NF1* mutations in 100% of the samples and a somatic *NF1* hit in 74% of cells from PNs, which was comparable with previous studies [25].

Biallelic *NF1* inactivation in benign dermal neurofibromas is a known phenomenon, and it is supposed that all lesional tissues develop from a second hit that may occur in utero or during the entire postnatal life. In the present patient’s superimposed mosaic involvement, however, the decisive difference is the period of time when the second hit occurs, which is most likely during the first week after fertilization [33,34]. Because all mosaic tissues represent a mixture of mutant and normal cells, it seems reasonable to assume that the presence of the second mutation in about 16% of examined cells was sufficient to express superimposed mosaicism in our patient. It is well known that superimposed mosaic NF1 lesions are particularly prone to develop malignant degeneration in the form of peripheral nerve sheath tumors (PNSTs) [23,27]. 

In summary, our results open up new avenues to explore the background of superimposed mosaic NF1. The use of whole exome sequencing allows a better clinical-molecular approach, leading to more focused therapy. In cases of segmental plexiform neurofibroma, future studies that take into account the molecular characterization we performed are required to identify the exact underlying pathomechanism.

## 4. Materials and Methods

### 4.1. NGS Based Gene Panel Sequencing of NF1, NF2, RAF1, KIT, SPRED1, SMARCB1, PTPN11 Genes

Molecular analysis of the DNA isolated from peripheral blood and plexiform neurofibroma of the proband was performed with a custom designed multi-gene panel assay examining *NF1*, *NF2*, *RAF1*, *KIT*, *SPRED1*, *SMARCB1*, and *PTPN11* genes. The libraries were prepared by using QIAGEN QIAseq targeted DNA custom panel kit (Qiagen, Hilden, Germany), and sequencing was performed on a MiSeq device (Illumina, San Diego, CA, USA) by using paired-end 150 bp reads. 

### 4.2. MLPA Analysis

Multiplex ligation-dependent probe amplification (MLPA) assays were used to screen for large deletions or duplications in the *NF1* gene by using the commercially available SALSA MLPA kits P081-D1 and P082-C2 (MRC-Holland, Amsterdam, The Netherlands). The two probe mixes contained one probe for each exon, three probes for exon 1, one probe for intron 1, and two probes for the exons 15, 21, 23, 51, and 58 of the NF1 gene. Additionally, one upstream and one downstream probe of *NF1* gene and two probes for the *OMG* gene (located within intron 36 of *NF1* gene) were applied. Following the manufacturer’s instructions, 100–200 ng of DNA was isolated from the peripheral blood sample and from the plexiform neurofibroma of the patient, and the same amount of three control genomic DNA was used for hybridization. Amplification products were separated by capillary electrophoresis on an ABI 3130 Genetic Analyzer (Life Technologies, Carlsbad, CA, USA), and the results were analyzed by using Coffalyser software (MRC-Holland, Amsterdam, The Netherlands). Each MLPA signal was normalized and compared with the corresponding peak area obtained from the three control samples. Deletions and duplications of the targeted regions were suspected when the signal ratio exceeded a 30% deviation.

### 4.3. Whole Exome Sequencing and Data Analysis

Whole exome sequencing was performed by using DNA samples obtained from unaffected skin, and peripheral blood samples and a second sample of plexiform neurofibroma. Exomic libraries were prepared by using the Illumina DNA Preparation with Enrichment Kit and sequencing was performed on an Illumina NovaSeq 6000 instrument with paired-end 100 bp reads. The mean sequencing depth of on target regions was 100.4X–123.5X. Base-called raw sequencing data was transformed into FASTQ format by using Illumina’s software (bcl2fastq). Reads were aligned to the human reference genome (GRCh37:hg19) by using Burrows–Wheeler Aligner. GATK algorithms (Sentieon Inc., San Jose, CA, USA) were used for duplicate read marking, local realignment around indels, base quality score recalibration, and variant calling. Sequencing depth and coverage were calculated based on the alignments. Sequence data for *NF1* gene were filtered for further analysis.

## 5. Conclusions

The clinical implications of our paper involve the potential for improved diagnosis and personalized treatment strategies through the identification of specific *NF1* gene mutations and the use of targeted therapies, such as selumetinib. These findings highlight the importance of incorporating molecular characterization into the clinical management of neurofibromatosis.

In terms of research implications, this case contributes to our understanding of the genetic basis and pathogenesis of segmental neurofibromas. The identification of an early second-hit somatic mutation suggests a potential mechanism underlying the extensive segmental form of the disease. Further research is needed to explore these molecular events and cellular pathways, providing insights for the development of future therapeutic targets.

Overall, this case report emphasizes the clinical and research value of genetic characterization and targeted therapies in neurofibromatosis, paving the way for improved patient care and advancing our understanding of the underlying mechanisms of plexiform neurofibromas.

## Figures and Tables

**Figure 1 ijms-24-12154-f001:**
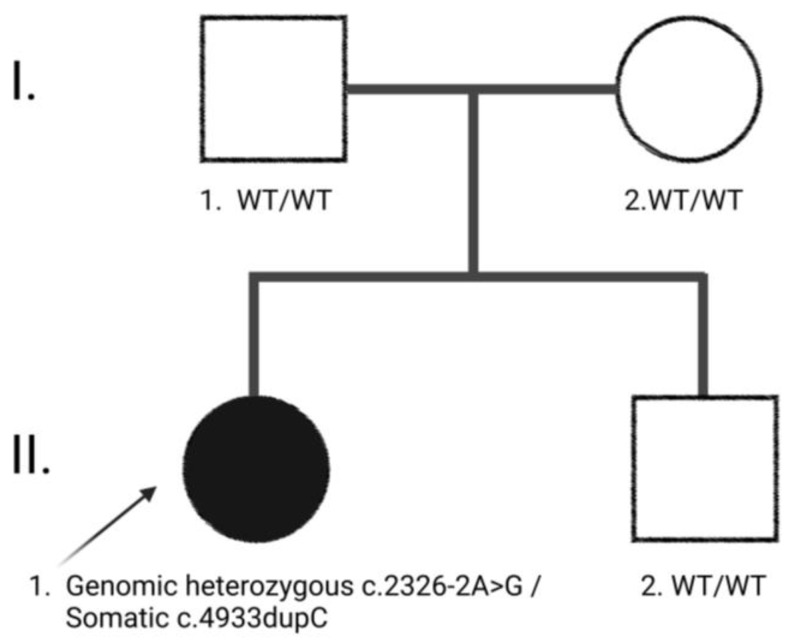
Family pedigree of the patient with *NF1* gene variants. Symbols and abbreviations used are denoted as follows: Arrow, index case (proband); Squares indicate males, circles indicate females, blackened symbol denote affected individual; WT: Wild Type.

**Figure 2 ijms-24-12154-f002:**
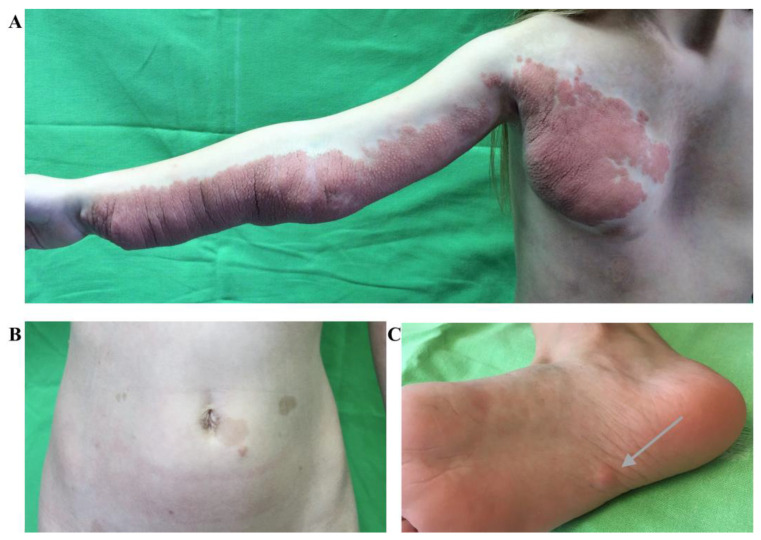
(**A**) Well-demarcated, brownish-livid soft tumor mass with several mobile, tender nodules on the medial side of the right arm from the palm to the axillary region with a segmental distribution, extending to the right side of the chest and pronounced pectus excavatum as a representation of plexiform neurofibroma. (**B**) Multiple, homogeneously hyperpigmented macules on the chest and trunk consistent with CALMs. (**C**) Small, soft, skin-colored cutaneous neurofibroma (arrow) on the sole.

**Figure 3 ijms-24-12154-f003:**
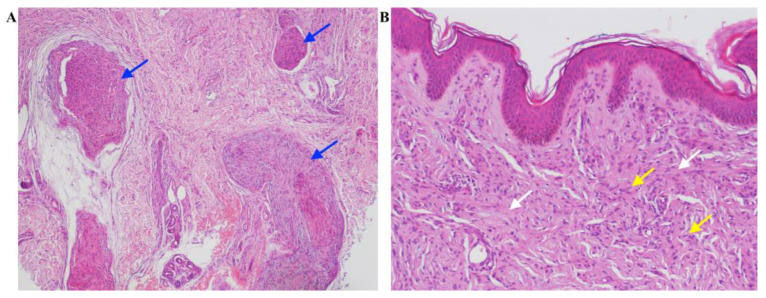
Histopathological specimens of the tumor mass on the right forearm. Non-encapsulated tumor mass in the dermis consisting of spindle cells with ovoid and wavy bland nuclei with pale eosinophilic cytoplasm (yellow arrows). Thickened nerve branches (blue arrows), collagen bundles (white arrows), and capillaries among the tumor cells. Histopathological findings of the samples are consistent with plexiform neurofibroma. (**A**) hematoxylin and eosin staining, magnification, ×40, (**B**) magnification, ×100.

**Figure 4 ijms-24-12154-f004:**
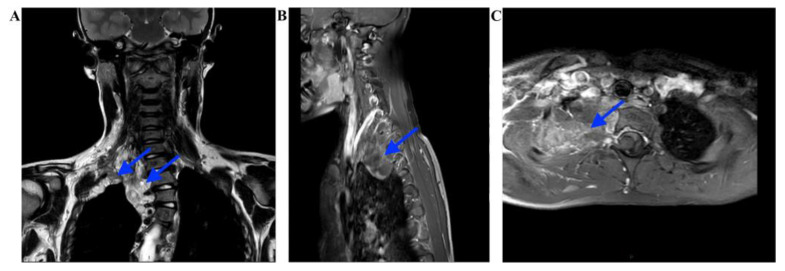
MRI scan of the chest shows a large heterogenous T2 hyperintense tumor (blue arrows) consisting of multiple nodules on the right side of the thoracic spine, extending from the lung apex to the upper mediastinum. Consequential left convex scoliosis is also seen. (**A**) T2 weighted, coroneal view, (**B**) contrast-enhanced T1, sagittal view, (**C**) and contrast-enhanced T1, axial view.

**Figure 5 ijms-24-12154-f005:**
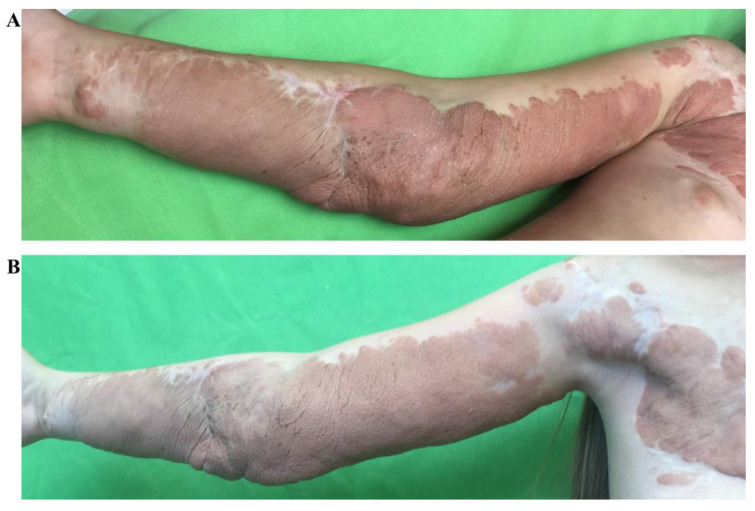
Selumetinib therapy response. The whole tumor mass became less tight, the tender nodules decreased from 30 to 10 mm, and her quality of life improved significantly. (**A**) Before treatment. (**B**) Six months after selumetinib initiation.

**Figure 6 ijms-24-12154-f006:**
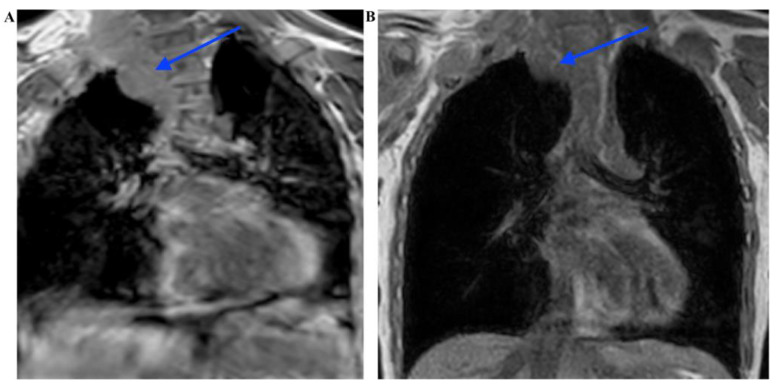
MRI image shows a reduction in size of the mediastinal tumor mass (blue arrows) up to 30%. MRI scan of the chest before treatment (**A**) and 4 months after the administration of selumetinib (**B**). T1 weighted, coroneal view.

**Figure 7 ijms-24-12154-f007:**
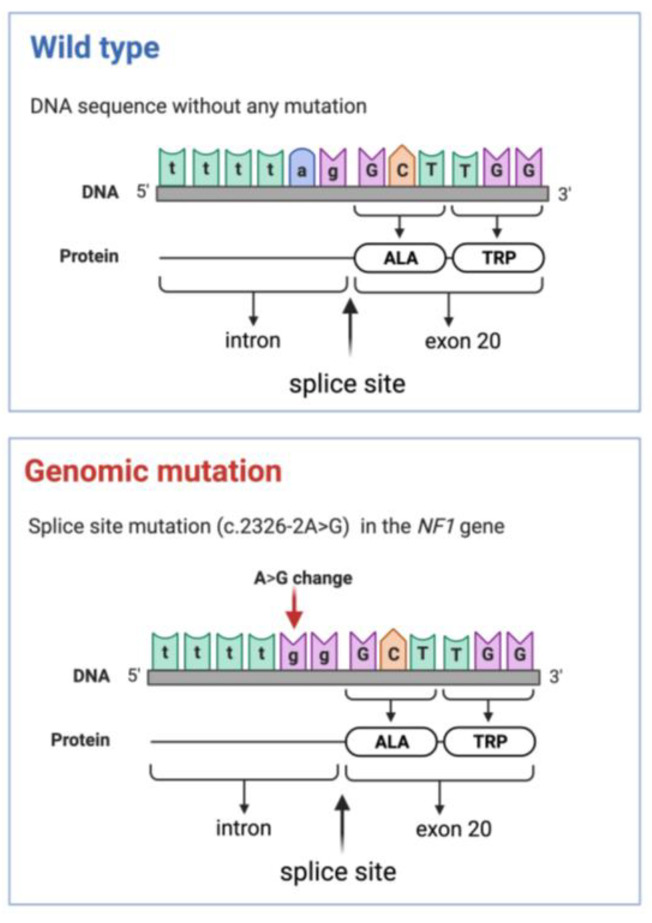
Schematic illustration of the genomic mutation in the *NF1* gene: splice-site mutation (c.2326-2A>G) in the intron before exon 20, compared to wild type. The figure was created with BioRender.com (accessed on 13 July 2023).

**Figure 8 ijms-24-12154-f008:**
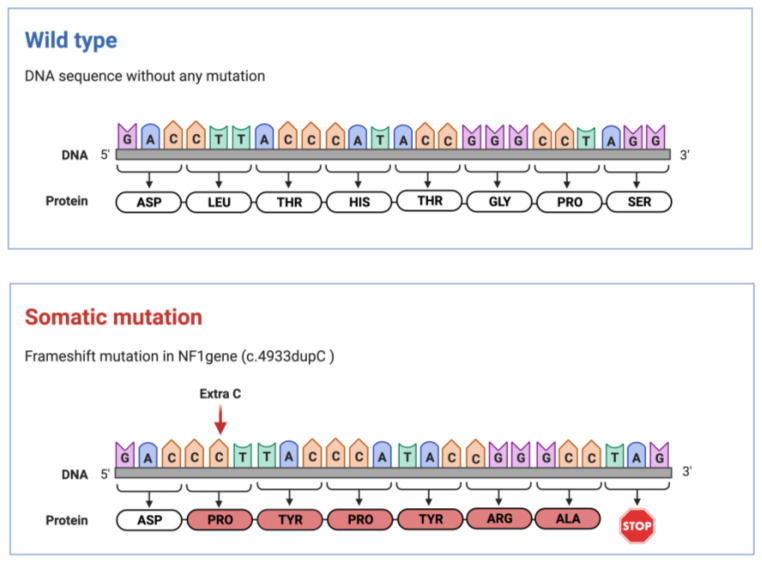
Schematic illustration of the somatic mutation in the *NF1* gene: a novel frameshift variant (c.4933dupC/p.Leu1645Profs*7/) exclusively in plexiform neurofibroma with a 16% variant allele frequency compared to wild type. The figure was created with BioRender.com (accessed on 13 July 2023).

## Data Availability

The data presented in this study are available on request from the corresponding author.

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
