# Peer review of "Superimposed Mosaicism in the Form of Extremely Extended Segmental Plexiform Neurofibroma Caused by a Novel Pathogenic Variant in the NF1 Gene"

_ijms, 2023, doi:10.3390/ijms241512154_

Round 1

Reviewer 1 Report

The aim of the authors' study was to reveal the role of superimposed mosaicism in the pathogenesis of plexiform neurofibroma in the NF1 gene region. Taking into account the frequency of neurofibroma detection and the role in the pathogenesis of mutations in the gene with varying degrees of manifestation, variants of the course and timing of detection of this pathology, low efficiency of traditional methods of treatment, the search for new potential points of impact on the mechanisms of formation (signaling pathways) - this study is very relevant. The approach used by the authors to find the cause of the genetic breakdown in the NF1 gene region is original, reproducible, but as the authors point out also has its shortcomings, which are due to technical development rather than the authors' ability to do so.During the study, the authors showed the existence of a new variant of genetic breakdown in the region of the FN1 gene and its significance in the clinical form of the disease. This is the main achievement of this study and can be used to further search for options to influence the pathology in order to regress it. The results of the research are documented and there is no doubt about the inferences made on their basis by the authors of the work. To the list of references there is a wish to use more modern literature, but this may also be a consequence of not so extensive research in this field.

Reviewer 2 Report

As a case report, there appears to be sufficient data related to the clinical treatment, the diagnosis, the status of the patient before and after treatment; and also the pathology of the patients tumors have been documented in sufficient detail. 

The documentation of the mutations identified, however, especially of the mosaicism described at the end of the manuscript, could benefit from more details. But maybe not necessarily in words: I have a hard time imagining the situation in the patients tumor cells: which cells carry which alleles, bot mutations are found with different frequencies, etc. I wonder if the authors could think of a way how this could be more easy to comprehend, especially for people that do not interpret genetic analyses on a daily bases. 

The article could benefit anyway from a "graphical abstract", maybe that could be killing 2 birds with one stone. 

Otherwise, there are no major concerns about the scientific soundness of the article, and the findings described. It is a case report; one does not expect functional analyses of the mutations effects on cell growth etc, or sensitivity to drugs. Its also rather short, because of this. Therefore, I think adding a graphical and schematic representation of the genetic findings themselves may benefit the paper. 

Reviewer 3 Report

1. Any genetic pedigree chart for this case?

2. Any IHC staining on BF1, KIT, RAF1, SPRED1, SMARCB1, and PTPN11 on Figure 2?

3. Provide a figure which summaries NF1 protein domains and the mutation site of this case.

Round 2

Reviewer 3 Report

No more comments